# Plasma Extracellular Vesicles from Preeclamptic Patients Trigger a Detrimental Crosstalk Between Glomerular Endothelial Cells and Podocytes Involving Endothelin-1

**DOI:** 10.3390/ijms26114962

**Published:** 2025-05-22

**Authors:** Elena Grossini, Marco Quaglia, Stefania Prenna, Alessandra Stasi, Rossana Franzin, Giuseppe Castellano, Valentino Remorgida, Alessandro Libretti, Sakthipriyan Venkatesan, Carlo Smirne, Guido Merlotti, Carmen Imma Aquino, Stefania Bruno, Giovanni Camussi, Daniela Surico, Vincenzo Cantaluppi

**Affiliations:** 1Laboratory of Physiology, Department of Translational Medicine (DIMET), University of Piemonte Orientale (UPO), 28100 Novara, Italy; elena.grossini@med.uniupo.it (E.G.); sakthipriyan.venkatesan@uniupo.it (S.V.); 2AGING Project Unit, Department of Translational Medicine (DIMET), University of Piemonte Orientale (UPO), 28100 Novara, Italy; marco.quaglia@med.uniupo.it (M.Q.); stefania.prenna@med.uniupo.it (S.P.); valentino.remorgida@med.uniupo.it (V.R.); alessandro.libretti@med.uniupo.it (A.L.); carlo.smirne@med.uniupo.it (C.S.); carmen.aquino@uniupo.it (C.I.A.); daniela.surico@med.uniupo.it (D.S.); 3Nephrology and Dialysis Unit, Department of Translational Medicine (DIMET), AOU S.S. Antonio e Biagio e Cesare Arrigo, University of Piemonte Orientale (UPO), 15121 Alessandria, Italy; 4Nephrology and Kidney Transplantation Unit, Department of Translational Medicine (DIMET), AOU Maggiore della Carità, University of Piemonte Orientale (UPO), 28100 Novara, Italy; 5Nephrology, Dialysis and Kidney Transplantation Unit, Department of Precision and Regenerative Medicine and Ionian Area (DiMePRe-J), University of Bari, 70124 Bari, Italy; alessandra.stasi@uniba.it (A.S.); rossana.franzin@uniba.it (R.F.); 6Nephrology, Dialysis and Kidney Transplantation Unit, Fondazione IRCCS Ca’ Granda Ospedale Maggiore Policlinico, Department of Clinical Sciences and Community Health, University of Milano, 20122 Milano, Italy; giuseppe.castellano@unimi.it; 7Obstetrics and Gynecology Unit, Department of Translational Medicine (DIMET), University of Piemonte Orientale (UPO), 28100 Novara, Italy; 8Internal Medicine Unit, Department of Translational Medicine (DIMET), University of Piemonte Orientale (UPO), 28100 Novara, Italy; 9Department of Primary Care, Azienda Socio Sanitaria Territoriale (ASST) of Pavia, 27100 Pavia, Italy; guido.merlotti@uniupo.it; 10Department of Medical Sciences, University of Torino, 10127 Torino, Italy; stefania.bruno@unito.it (S.B.); giovanni.camussi@unito.it (G.C.)

**Keywords:** preeclampsia, extracellular vesicles, endothelial dysfunction, proteinuria, glomerular injury, Endothelin-1

## Abstract

Extracellular vesicles (EVs) may play a role in preeclampsia (PE)-associated glomerular damage. We herein investigated the role of PE plasma EVs in triggering a detrimental crosstalk between glomerular endothelial cells (GEC) and podocytes (PODO). Clinical and laboratory variables were examined at T0 (diagnosis), T1 (delivery), and T2 (one month after delivery) in 36 PE patients and 17 age-matched controls. NanoSight and MACSPlex evaluated EV concentration, size, and phenotype. GEC and PODO were stimulated with plasma EVs to study viability, reactive oxygen species (ROS) production, permeability to albumin, endothelial-to-mesenchymal transition, and Endothelin-1 release. EV size and concentration were higher in PE than in healthy controls and in severe than in mild forms of disease. At T0, higher EV concentration correlated with proteinuria, blood pressure, uric acid, and liver enzyme levels. PE-EVs originated from leukocytes, endothelial cells, platelets, and the placenta and induced GEC and PODO damage as shown by the reduction of viability, increased ROS release, and albumin permeability. Co-culture experiments demonstrated that PE-EVs mediated a deleterious intraglomerular crosstalk through Endothelin-1 release from GEC able to down-regulate nephrin in PODO. In conclusion, we observed in PE plasma a peculiar pattern of EVs able to affect GEC and PODO functions and to induce proteinuria through Endothelin-1 involvement.

## 1. Introduction

Preeclampsia (PE) is a pregnancy disorder associated with new-onset hypertension, which occurs most often after 20 weeks of gestation, often with accompanying proteinuria [1]. It is a frequent cause of maternal death and of complications for newborns [2,3]. PE risk factors include history of PE, autoimmune diseases (such as Systemic Lupus Eritematosus), diabetes mellitus, chronic kidney disease, multifetal gestation, and in vitro fertilization [1,4]. The initial trigger of PE is represented by a placentation defect which results in an altered perfusion with consequent tissue hypoxia, ischemic damage, syncytiotrophoblast (STB) apoptosis and the release of antiangiogenic and vasoconstrictive factors, hypoxia-inducible factor-α (HIF-α), reactive oxygen species (ROS), and inflammatory cytokines into the maternal circulation [5,6]. These alterations lead to the second stage of PE characterized by systemic endothelial dysfunction and organ damage, including that of the kidney, with the development of proteinuria and progression toward chronic kidney disease (CKD) [6,7,8].

Among circulating factors present in PE, extracellular vesicles (EVs) may play a crucial role [9], being involved in migration and the invasion of trophoblasts and cellular adaptations to physiological changes [10,11,12,13]. EVs are small microparticles mainly belonging to the exosome and shedding vesicle (microvesicles) families involved in cell-to-cell communication through the horizontal transfer of proteins, receptors, lipids, and genetic material [14]. Recent studies have focused on EVs derived from the maternal vasculature and placental syncytiotrophoblast (STB). Of note, changes of EV concentration and phenotype may contribute to PE pathophysiology by enhancing the pro-inflammatory and pro-coagulant state of pregnancy [15], thus contributing to endothelial dysfunction, which precedes the increase of vascular resistance [16]. However, few studies have focused on the relevance of plasma EVs as mediators of PE-associated glomerular functional alterations and damage. Our group previously showed that a PE-associated increase of glomerular permeability and proteinuria may be explained by nephrin loss in podocytes (PODO) and that circulating factors contained in PE sera stimulated glomerular endothelial cells (GEC) to increase the release of Endothelin-1 (ET-1) that, in turn, is responsible for nephrin shedding from PODO [17].

The aims of this study were as follows: (1) to evaluate the pattern of plasma EVs in PE patients at different time points and to correlate their phenotype with the severity of disease and clinical variables and (2) to investigate in vitro the potential role of EVs in the pathogenic mechanisms of PE by performing experiments on GEC and PODO separately or in a co-culture model mimicking the glomerular filtration barrier.

## 2. Results

### 2.1. Demographic and Clinical Profiling of PE Patients

We enrolled a total of 53 patients, among which 36 were diagnosed with PE (18 severe and 18 mild forms) and 17 were Healthy Controls (HCs); the study took place in our hospital that manages about 1850 annual deliveries/year, with an average 5% of PE cases/year. The demographic data of the enrolled patients are reported in Table 1.

Severe PE was characterized by a higher BMI, and previous smokers were more represented. Among all PE patients, 4/36 (11.1%) had drug-treated hypertension and 12/36 (33.3%) had previous complicated pregnancies. The mean gestational age was 33 weeks in severe PE, 37 in mild PE, and 37 in HCs (Table 2).

Among patients with severe PE, 5/18 (27.7%) developed HELLP syndrome. Almost all deliveries before 34 weeks belonged to the severe PE group, whereas in almost all HCs, they occurred after 39 weeks. All newborns weighing less than 1 kg and 64% of those weighing 1–2.5 kg were associated with severe PE. In addition, 70% of hospitalizations in the neonatal intensive care unit (NICU) were due to the presence of severe PE (*p* = 0.002). The clinical and laboratory variables measured at T0 (study enrollment) in PE and in HCs matched for gestational age are described in Table 3.

Liver enzymes > 100 U/L, platelet count < 100 × 10^3^, and uric acid > 6 mg/dL were more frequent in the PE than in the HC group; uric acid levels higher than 6 mg/dL were found exclusively in PE patients (Table 4).

No difference was observed in proteinuria between mild and severe PE. Two severe PE patients showed ARED (Absent or Reverse End Diastolic) in the umbilical artery. Moreover, a notch of uterine blood flow was found only in 6 PE patients. At T2, SAP and DAP returned to normal values in almost all PE patients, whereas about half of them showed persistent proteinuria > 0.6 g/day.

### 2.2. Phenotypic Characterization of Plasma EVs and Correlation with Clinical Parameters

NanoSight analysis performed at T0 and T1 showed that both size and concentration of plasma EVs were higher in severe PE than in mild PE and HCs (Figure 1). At T2, EV size in both severe and mild PE forms remained higher than HC, whereas no significant difference was observed for EV concentrations (Figure 1).

Characterization of the EVs isolated from PE patients showed leukocytes (T and B lymphocytes in Figure 2A–E and monocytes in Figure 2F), endothelial cells (Figure 2G), and platelets (Figure 2H) as the main sources. The expression of these specific cell markers changed over the study period: an enhanced expression of lymphocyte, endothelial and platelet markers was detected in PE rather than in HCs at T2 (Figure 2A–H). Significant differences in these markers were also observed between severe and mild PE patients at all time points. Moreover, a higher expression of the placental antigen PLAP (Figure 2I) and of TF involved in coagulation (Figure 2J) was found in PE EVs at all time points considered, predominantly in the most severe forms. The exosomal markers CD9, CD63, and CD81 were used as a control for flow cytometry studies [18]. A positive correlation was also found between EV concentration and several clinical variables in PE patients at T0, including the values of SAP (Pearson r: 0.84; *p* < 0.0001), DAP (Pearson r: 0.81; *p* < 0.0001), the Pulsatility Index (PI) of the umbilical arteries (Pearson r: 0.77; *p* < 0.0001), and, in particular, proteinuria (Pearson r: 0.89; *p* < 0.0001). Significant correlations were also shown for liver enzymes AST (Pearson r: 0.79; *p* < 0.0001) and ALT (Pearson r: 0.70; *p* = 0.0005), uric acid (Pearson r: 0.75; *p* < 0.0001), and for the anti-angiogenic factor s-Flt1 (Pearson r: 0.99; *p* < 0.0001).

### 2.3. In Vitro Experiments on GEC and PODO Directly Stimulated by Plasma EVs

The incubation of GEC and PODO with PE EVs caused a reduction of cell viability (Figure 3A,B) and an increase of ROS release (Figure 3C,D) in both cell lines throughout T0 to T2. This cytotoxic effect was more pronounced in the severe than in the mild PE forms. Of note, cell viability at T2 was reduced in GEC treated with EVs from both mild and severe PE, whereas it was reduced in PODO only with EVs from severe PE (Figure 3A,B). Furthermore, nitric oxide (NO) release was higher in GEC stimulated with EVs from severe, rather than mild, PE at T0 and from both severe and mild PE, rather than HCs, at T2 (Figure 3E).

Incubation of GEC with PE EVs was also able to induce Endothelial-to-Mesenchymal Transition (EndMT) from T0 to T2. Hence, CD31 and VE-cadherin expressions were reduced at T0 and T1 in PE patients (Figure 4A,B). Of note, although both CD31 and VE-cadherin expression increased at T2 as compared with previous time points, their levels were still lower than that of HCs (Figure 4A,B). The reduced expression of endothelial markers was concomitant to the up-regulation of type 1 collagen and vimentin at T0 (Figure 4C,D). In severe PE, the expression of type 1 collagen 1 and vimentin was still higher than that observed in HCs at both T1 and T2, whereas in mild PE this was observed only for type 1 collagen (Figure 4C,D). EndMT was associated with an altered function of GEC, as confirmed by the albumin diffusion test and analysis of Endothelin-1 release. Using GEC supernatants, albumin diffusion was higher than that observed with the supernatant from HCs at T1 and T2 and it was higher in severe PE than in the mild forms of PE at all time points (Figure 4E). Similarly, ET-1 release was increased in GEC treated with EVs from severe PE patients vs. HCs at all time points. Moreover, ET-1 levels were significantly higher in severe than in mild forms of PE. Of note, ET-1 release remained significantly higher than in HCs also at T2 (Figure 4F).

PE EVs also determined PODO damage, confirmed by altered albumin permeability and nephrin shedding. As described for GEC, albumin diffusion was also higher in severe rather than mild PE at T0 and T2, and in PE vs. HCs at T1 and T2 (Figure 5A). Nephrin expression was also significantly reduced in PODO, particularly in severe PE at T0 (Figure 5B). Although a partial recovery was observed at T1 and T2, it was still significantly lower in severe PE than in HCs (Figure 5B). Furthermore, the release of VEGF-A from PODO treated with PE EVs was reduced in comparison to HCs, and it was lower in severe rather than in mild PE forms at all time points (Figure 5C).

### 2.4. GEC-PODO Co-Culture Experiments

The evidence of plasma PE EV-induced release of ET-1 by GEC prompted us to investigate the presence of a possible detrimental crosstalk sustained by this mediator in the glomerular microenvironment. For this purpose, we developed a GEC-PODO co-culture method in which one cell type was stimulated with the supernatant of the other cell line after challenging it with plasma EVs from PE patients or HCs. Supernatants from PE EV-stimulated PODO induced in GEC a significant decrease of viability (Figure 6A) and ROS release (Figure 6B) at all time points considered; likewise, supernatants from PE-EV-stimulated GEC induced in PODO a significant decrease of viability (Figure 6C) and ROS release (Figure 6D). Similar results were observed for albumin diffusion in GEC (Figure 6E) and PODO (Figure 6F). Moreover, incubation with supernatants from PE EV-stimulated GEC down-regulated nephrin expression in PODO at all time points (Figure 6G). Of note, the results obtained with GEC or PODO supernatants were comparable to those obtained with the direct stimulation of both cell lines with plasma PE EVs. This detrimental crosstalk was not observed using supernatants collected after stimulation of GEC or PODO with plasma EVs collected from HCs (Figure 6A–G). Last, the alterations of albumin diffusion and nephrin expression in PODO after incubation with supernatants collected from plasma PE EV-stimulated GEC were significantly inhibited at all time points by the presence of the ET-1 receptor antagonist PD142893, suggesting a key pathogenic role of this molecule in PODO injury (Figure 6F,G).

## 3. Discussion

In this study, we described the role of plasma EV as both biomarkers and mediators of PE-associated glomerular damage. We found that the size and concentration of EVs were higher in PE as compared to normal pregnancies, particularly in the most severe forms of disease. The EV phenotype of PE patients showed the presence of antigens of leukocytes (in particular lymphocytes and monocytes), platelets, endothelial cells, and placental cells, along with biomarkers of exosomes and of activation of the inflammatory and coagulative cascades. Moreover, PE plasma EV concentrations correlated with proteinuria, hypertension, elevated serum levels of uric acid, and liver enzymes. We also evaluated in vitro the biological effects of EVs on human GEC and PODO. We observed that PE-EVs directly damaged both glomerular cell types, causing decreased viability, increased ROS generation and alterations of albumin permeability. Co-culture experiments confirmed that also the supernatants derived from GEC or PODO after the plasma PE EV challenge were able to induce functional alterations and damage in the other cell line. In particular, we found that ET-1 released by plasma PE EV-stimulated GEC down-regulated nephrin expression in PODO. Taken together, these results suggest a key role for plasma EVs in triggering a detrimental crosstalk between GEC and PODO, leading to PE-associated glomerular injury and proteinuria.

The first relevant finding of this study was that EV concentration was higher in PE patients than in HCs at T1 and T2, and severe PE patients had a higher concentration of EVs than mild ones at all time points considered, suggesting that the inflammatory reaction and the cellular damage associated with PE may enhance the release of EV that could be used as biomarkers of disease state and activity. It was also of interest that EV size was higher in severe PE than in mild PE at T0. Unlike EV concentration that decreased at T2 in PE patients, EV size remained larger than that of HCs also at this point, suggesting a persistently abnormal morphology for at least three months post-partum. Moreover, the EV phenotype was different among the groups: a percentage of PE-EVs isolated at T0 showed the presence of PLAP [19], suggesting a placental origin [20]. This marker decreased over time from T0 to post-partum, but remained higher in PE patients than in HCs at T1 and T2. PLAP positivity was also stronger in severe rather than in mild forms of PE at T0 and T1. These results are consistent with a key role of placental EV in mechanisms triggering PE, as, in this setting, the STB releases abnormal EVs, together with other anti-angiogenetic factors such as sFlt-1 [21,22,23]. However, the surface EV phenotype also showed the presence of markers of endothelial cells, platelets, and lymphocytes, indicating a peculiar profile of derivation of PE-EVs, possibly reflecting key mechanisms of disease, such as endothelial dysfunction, platelets, and immune system activation.

In the last decade, EVs have extended their relevance to different fields of medical research for their crucial role in the mechanisms of cell-to-cell communication. EVs are formed by two main families: exosomes that have a size of 30–150 nm and which are generated by multivesicular bodies and shedding vesicles or microvesicles, which are larger in size and are released by activated cells through a membrane-sorting process involving different calcium-dependent enzymes [24]. In the clinical scenario of inflammatory diseases, our group has recently demonstrated that plasma EVs isolated from chronic hemodialysis patients express surface markers involved in inflammation and activation of the complement and coagulation cascades [25].

EVs play a crucial role in physiological pregnancy through the modulation of embryo implantation, trophoblast invasion, maternal immune responses, and spiral arterioles remodeling [26,27,28]. Beneficial or deleterious effects of EVs could depend on their molecular phenotype and concentration: changes in the EV profile could shift the balance towards the development of disease. This occurrence can also be observed in conditions of tissue hypoperfusion, inflammation, and oxidative stress, which are similar to the typical microenvironment observed in PE [29].

Another interesting aspect of this study is the relationship between EV concentration and sFLT-1, an established biomarker of PE. Soluble FLT-1 is a key anti-angiogenic factor, together with Endoglin (sEng), and both are increased in the plasma of PE patients [30,31,32]. These mediators contribute to endothelial dysfunction and systemic inflammation by counteracting pro-angiogenic molecules, such as placental growth factor (PIGF) and VEGF. As expected, we found elevated plasma levels of sFlt-1 in PE patients, both at T0 and T1, and higher levels in severe than in mild forms. Of interest, we also found a positive correlation between sFlt-1 and EV concentration at T0. In addition, EVs correlated with clinical and laboratory parameters such as proteinuria, uric acid, and liver enzyme concentration, the PI of umbilical arteries, and SAP/DAP values. Placental EVs may modulate lipid metabolism in the liver through direct interaction with hepatocytes [33]. Moreover, the correlation with serum uric acid, a known early predictor of PE and a marker of cardiovascular and CKD risk, is also consistent with their pathogenic role [34,35,36].

In the second part of the study, the relationship between EVs and glomerular injury in PE was confirmed by in vitro experiments. The treatment of GEC and PODO with PE-EVs reduced their viability and increased ROS release and albumin permeability in both cell types. PE-EVs induced nephrin shedding from PODO and EndMT in GEC, a process characterized by the loss of endothelial markers and by the acquirement of a fibroblast-like phenotype able to trigger tissue fibrosis [37]. Although a general improvement was observed at T2, particularly in severe patients, almost all cellular variables remained altered and differed from those observed in HCs. Evidence of the persistent effects of PE-EVs of inflammatory and endothelial origin even after delivery may suggest that they could be involved in long-term systemic sequelae, particularly an increased cardiovascular risk.

One of the most interesting findings is the role of PE-EVs in eliciting the release of critical mediators (ET-1 and VEGF) by GEC and PODO, modulating the crosstalk between these two cell types and switching it toward pathology. The in vitro effects on GEC and PODO in co-culture were similar to those observed with the direct stimulation of single cell lines with PE-EVs. These findings suggest that EVs may induce the release of paracrine mediators from GEC, which, in turn, acts on PODO, reproducing detrimental effects like those observed with the separate stimulation of PODO with EVs. In the presence of the Endothelin-1 receptor antagonist PD142893, these deleterious effects were significantly blunted, suggesting a key role of this mediator as an effector of PODO damage triggered by a dysfunctional endothelial cell phenotype. On the other hand, the release of VEGF from PODO treated with PE-EVs was reduced in comparison with HCs and was lower in severe than in mild PE at all time points. Taken together, these results provide some new possible elements in the complicated crosstalk between GEC and PODO in PE. Recently, Collino et al. [17] demonstrated that serum from PE patients did not directly downregulate nephrin expression in PODO, whereas conditioned medium obtained from GEC after incubation with PE sera induced nephrin shedding from the cell surface. Moreover, an Endothelin-1 receptor antagonist abrogated these effects, whereas the administration of recombinant Endothelin-1 enhanced nephrin shedding. Furthermore, conditioned medium obtained from these cells determined nephrin loss. This study also clarified a mechanism probably linking endothelial injury from VEGF deficiency to altered PODO permeability. Other studies have shown that VEGF is constitutively released from PODO; it binds to endothelial cell receptors in a paracrine manner and is essential for GEC function and survival [38]. On the other hand, a dysfunctional endothelial cell can determine PODO injury [39,40]. Thus, disruption between GEC and PODO physiological crosstalk appears to be mediated by this vicious circle of mutual cell damage, with a pivotal role played by VEGF and Endothelin-1 [41,42].

Our study added some further elements to this complex scenario, emphasizing the role of PE EVs in increasing the release of ET-1 from GEC and concomitantly inhibiting that of VEGF from PODO. Previously observed effects of preeclamptic serum on GEC and PODO can be reproduced by PE-EVs alone. The effectiveness of the ET-1 blockade suggests that this approach may result not only in the treatment of the systemic effects associated with PE (e.g., arterial hypertension), but also in protection from PODO injury, directly interacting with mechanisms of proteinuria [43]. This is in accordance with recent preclinical data and RCTs showing a beneficial effect of selective endothelin receptor antagonists on CKD progression due to the limitation of vasoconstriction, inflammation, and fibrotic processes [44]. In addition to pharmacological approaches such as ET-1 blockade, previous clinical studies have demonstrated that therapeutic plasma exchange (TPE) during early-onset pre-eclampsia could improve pregnancy duration and reduce serum levels of antiangiogenic factors including sFlt-1 and endoglin [45]. Furthermore, it has been recently demonstrated that sFlt-1 and PlGF associate with both plasma lipoproteins and EVs [46]. Schroder et al. showed that different modalities of lipoprotein apheresis are able to remove EVs from the plasma of hypercholesterolemic patients at high risk of developing cardiovascular diseases [47].

We acknowledge that our study has some limitations: the limited sample size prevents us from drawing definite conclusions. The lack of a control group of healthy pregnancies with a comparable gestational age at diagnosis and relatively short follow-up (3 months) after pregnancy limits our analysis on the pathogenic role of PE EVs. Some future perspectives can be envisaged for studies on EVs in the setting of PE; it would be interesting to extend the follow-up of the patients and of the newborns after delivery to assess whether EV alterations persist or whether they return to a normal homeostasis. Prolonged abnormal levels or phenotype of EVs after delivery could potentially help to identify a subset of patients with persistent alterations of endothelial function and increased cardiovascular risk. These data could be useful to better characterize the role of EVs as new therapeutic targets for the prevention and treatment of PE, extending this concept after delivery [48]. A prospective study could identify the presence of abnormal EVs before the clinical onset of PE and thus assess their role as a very early marker of disease. A further development could be an analysis of EVs’ bioactive content, with the aim of characterizing especially miRNAs, which mediate most effects on target cells [49]. Recent studies have highlighted a potential role of different miRNAs, such as has-miR-9-5p as a biomarker of PE [50], miR-18b as a protective factor in PE patients [51], and miR-1273d, miR-4492, and miR-4417 as pathogenic factors [49]. A potential role of miRNA transfer by EVs in the modulation of ET-1 levels may represent a future point in the research agenda, since, to our knowledge, there are no data available in the literature.

## 4. Materials and Methods

### 4.1. Patients and Clinical Variables

PE patients were enrolled at the Gynecology and Obstetrics Unit, University of Piemonte Orientale (UPO), “Maggiore della Carità” University Hospital in Novara (Italy). The inclusion criteria were age > 18 years; gestational hypertension, PE, PE superimposed on chronic hypertension; and HELLP syndrome. Patients were sub-grouped into severe and mild PE forms, respectively, based on the presence or absence of systolic arterial blood pressure (SAP) > 160 mmHg or diastolic arterial blood pressure (DAP) > 110 mmHg; thrombocytopenia (<100,000 PTL/microL); serum creatinine > 1.2 mg/dl; the doubling of plasma liver enzymes; respiratory distress syndrome or pulmonary edema; and neurologic symptoms. Healthy controls (HCs) were age- and gestational age-matched with PE patients. The study was approved by the local Ethical Committee (Protocol n 26593 “Maggiore della Carità” University Hospital Novara, Study n CE172/17). Patients were informed of the research protocol and objectives and signed a written consent form. All patients were treated according to Good Clinical Practice standards [52]. SAP, DAP, liver enzymes, uric acid, WBC, RBC, platelet count, renal function, urine sediment, and 24 h proteinuria were examined at diagnosis (T0), delivery (T1), and at one month after delivery (T2). At different time points, plasma Fms-like tyrosine kinase 1 (sFlt-1) levels were also assessed by ELISA (Thermo Fisher Scientific, Monza, Italy) according to the manufacturer’s instructions.

### 4.2. Plasma EV Isolation and Characterization

Blood samples were taken from PE patients and HCs at different time points and were immediately centrifuged at 3000 rpm for 10 min at 4 °C. Plasma was then divided into 5 different tubes to avoid multiple thawing. EVs were isolated by plasma samples by serial centrifugation steps and their enrichment was finally obtained by ultracentrifugation for 2 h at 100,000× *g* at 4 °C using the SW60Ti rotor in a Beckman Coulter Optima L-90 K ultracentrifuge (Beckman Coulter, Fullerton, CA, USA). After ultracentrifugation, the supernatants were removed and the pellets re-suspended in DMEM supplemented with 1% DMSO and stored at −80 °C until use. The isolated EVs were diluted 1:1000 in sterile saline (BBraun, Milan, Italy) and analyzed by Nanosight (NS300 Malvern Panalytical, Malvern, UK) equipped with the Nanoparticle Tracking Analysis (NTA) software (NS300 Malvern) to determine the EV microparticle concentration and size. The total number of EVs for each patient was obtained by multiplying the value given by the instrument (microparticles/mL) for the dilution used for the analytic purpose, including the number of mL in which EVs were resuspended. The antigen surface expression of isolated EVs was studied by FACS using the commercially available MACSPlex kit (Miltenyi Biotec S.r.l., Bologna, Italy) that allows the detection of the expression of 39 surface markers by different antibody-coated bead subsets. Briefly, 1 × 10^10^ EVs were resuspended in the MACSPlex Buffer with 15 µL of the antibody-coated MACSPlex Exosome Capture Beads and 15 µL of MACSPlex Exosome Detection Reagent cocktail in accordance with the manufacturer’s instructions. After incubation for 1 h in the dark, 500 µL of MACSPlex Buffer was added and centrifuged at 3000× *g* for 5 min. The supernatant was then removed and this step was repeated twice. Each sample was analyzed using the Cytoflex flow cytometer (Beckman Coulter), whereby approximately 5000–8000 single bead events were recorded per sample. The median fluorescence intensity (MFI) for all 39 exosomal markers was corrected for background and gated based on the respective fluorescence intensity according to the manufacturer’s instructions. For other specific EV staining, the EVs were subjected to FACS analysis using antibodies directed to Placental Alkaline Phosphatase (PLAP) (Santa Cruz Biotechnologies, Santa Cruz, CA, USA) or Tissue Factor (TF) (Thermo Fisher Scientific). All the reported isolation and characterization procedures were performed in accordance with the International Society for Extracellular Vesicles (ISEV) guidelines and with previous published studies [53,54].

### 4.3. In Vitro Studies

Human Glomerular Endothelial Cells (GEC) and podocytes (PODO) were isolated and characterized as previously described [55,56,57]. After a direct stimulation with plasma-derived EVs, different experiments were performed on GEC cell viability using MTT (Life Technologies, Monza, Italy); ROS release (Abcam, Cambridge, UK); NO production by Griess method (Promega, Milan, Italy); FITC-albumin diffusion; ELISA for Endothelin-1 release (R&D Systems, Minneapolis, MN, USA); and endothelial-to-mesenchymal transition by FACS using specific conjugated antibodies directed to CD31, VE-cadherin, type 1 collagen or vimentin (all from Vinci Biochem, Firenze, Italy). Similar experiments were performed on PODO cell viability using MTT; ROS release; FITC-albumin diffusion; ELISA for VEGF release (R&D Systems); and nephrin expression by FACS using a specific FITC-conjugated antibody (Santa Cruz Biotechnology, Santa Cruz, CA, USA) [57,58,59,60,61,62,63,64,65]. In selected experiments, we developed a GEC-PODO co-culture model in which one cell type was stimulated with the supernatant produced by the other cell line, evaluating the same assays described above. Moreover, the role of the Endothelin-1 receptor antagonist PD142893 (10 µM) on nephrin expression and permeability to the albumin of PODO stimulated with GEC supernatants in the co-culture model was also investigated. Detailed information on in vitro studies on GEC and PODO are reported in the Appendix A.

### 4.4. Statistical Analysis

Statistical analysis was performed by GraphPad Prism 6 (San Diego, CA, USA). Demographic and clinical variables were expressed as mean, range, and standard deviation (SD). For categorical variables, data were checked for normality before statistical analysis. All results obtained were examined through a Mann–Whitney test. Correlation analysis was performed by using the Pearson correlation coefficient. All data are presented as dot plots, means, and ± 1SD of repeated measurements. A value of *p* <0.05 was considered statistically significant.

## 5. Conclusions

The results of the present study highlight the existence of a peculiar pattern of plasma EVs derived from endothelial cells, platelets, leukocytes, and the placenta, which correlate with the presence and the clinical severity of PE. The pathogenic role of PE-EVs was further confirmed by in vitro experiments showing their multiple biological effects on GEC and PODO, such as the reduction of cell viability, the increase of ROS release and albumin permeability, and the triggering of EndMT. Co-culture experiments suggest that PE-EVs may amplify a deleterious crosstalk between GEC and PODO by modulating the release of ET-1 and VEGF. Overall, EV study holds promise as a tool to expand our knowledge on the pathophysiology of PE and to discover new therapeutic targets to improve the management of this complex pregnancy-related disorder.

## Figures and Tables

**Figure 1 ijms-26-04962-f001:**
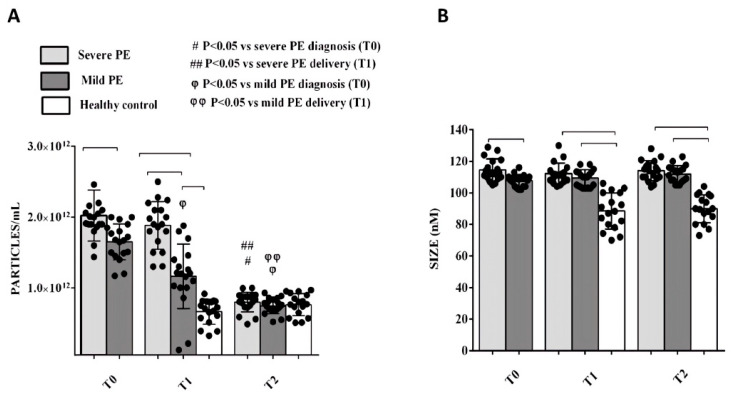
Plasma EV concentration (**A**) and size (**B**) in PE patients and HCs. T0: diagnosis; T1: delivery: T2: 1 month after delivery. Reported data are shown as dot plots and means ± 1SD of 5 different measurements. Square brackets indicate significance between groups (*p* < 0.05). Other significances are shown through symbols.

**Figure 2 ijms-26-04962-f002:**
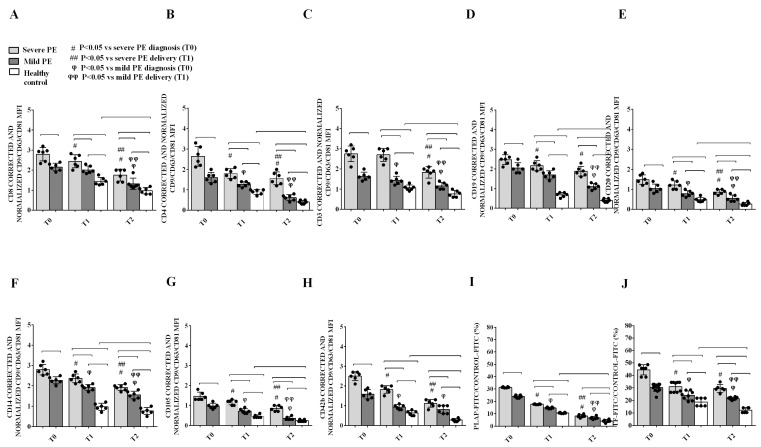
Expression of Lymphocyte (CD8, CD4, CD3, CD19, and CD20 (**A**–**E**), respectively), Monocyte (CD14 in (**F**)), Endothelial (CD105 in (**G**)) or Platelet (CD42b in (**H**)) biomarkers, Placental Alkaline Phosphatase (PLAP in (**I**)) and Tissue Factor (TF in (**J**)) in EVs from PE patients and HCs. T0: diagnosis; T1: delivery: T2: 1 month after delivery. MFI: median fluorescence intensity is also shown. T0: diagnosis; T1: delivery: T2: 1 month after delivery. MFI: median fluorescence intensity. Reported data are shown as dot plots and means ± 1SD of 5 different measurements. Square brackets indicate significance between groups (*p* < 0.05). Other significances are shown through symbols.

**Figure 3 ijms-26-04962-f003:**
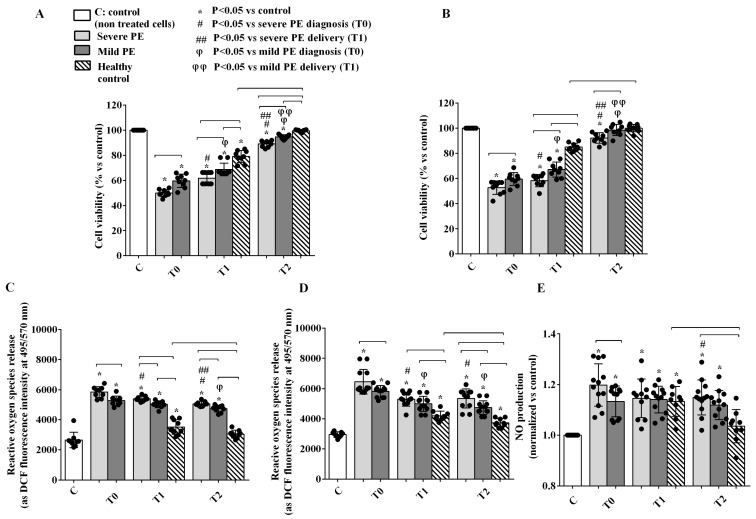
Effects of EVs from PE patients and HCs on cell viability and ROS production, respectively, in GEC (**A**,**C**) and PODO (**B**,**D**) or on NO release in GEC (**E**). Reported data are shown as dot plots and means ± 1SD of 5 different measurements. Square brackets indicate significance between groups (*p* < 0.05). Other significances are shown through symbols.

**Figure 4 ijms-26-04962-f004:**
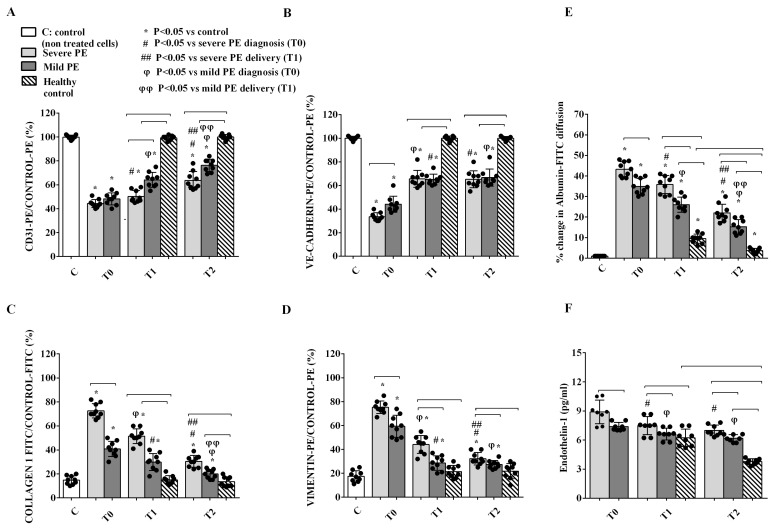
Endothelial-to-Mesenchymal Transition (EndMT) defined by expression of the endothelial antigens CD31 (**A**) or VE-cadherin (**B**) and of the fibroblast markers type I collagen (**C**) or vimentin (**D**), albumin diffusion, (**E**) and endothelin 1 release (**F**) in GEC treated with EVs from PE patients and HCs. T0: diagnosis; T1: delivery: T2: 1 month after delivery. Reported data are shown as dot plots and means ± 1SD of 5 different measurements. Square brackets indicate significance between groups (*p* < 0.05). Other significances are shown through symbols.

**Figure 5 ijms-26-04962-f005:**
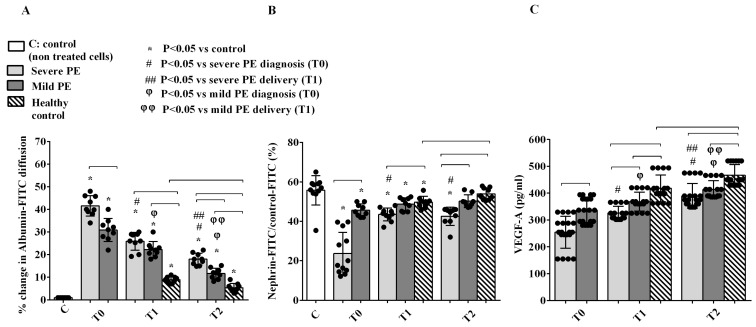
Effects of EVs from PE patients and HCs on albumin diffusion (**A**), nephrin expression (**B**) and VEGF-A release (**C**) in PODO. T0: diagnosis; T1: delivery: T2: 1 month after delivery. Reported data are shown as dot plots and means ± 1SD of 5 different measurements. Square brackets indicate significance between groups (*p* < 0.05). Other significances are shown through symbols.

**Figure 6 ijms-26-04962-f006:**
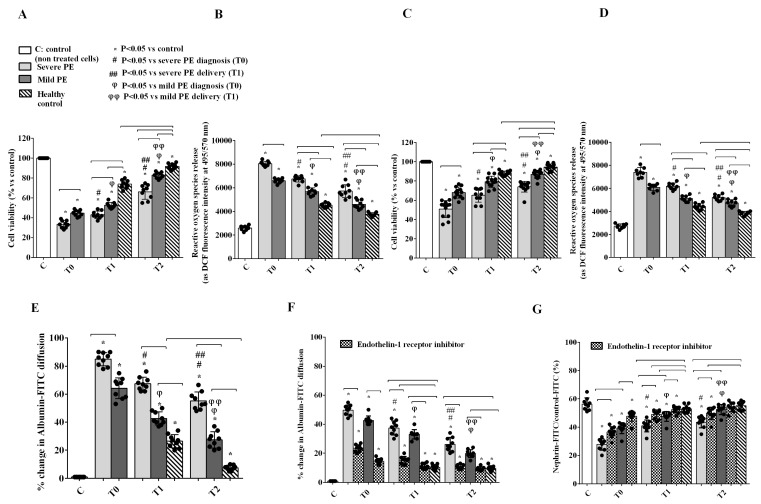
Effects of EVs from PE patients and healthy controls on cell viability (**A**,**C**) and ROS release (**B**,**D**) in GEC (**A**,**B**) and PODO (**C**,**D**) in a co-culture model. Albumin diffusion across multicellular layer of GEC (**E**) or PODO (**F**) and nephrin expression by PODO (**G**) in the same co-culture model. In (**F**,**G**), a specific column indicating the Endothelin-1 receptor inhibitor is also reported. T0: diagnosis; T1: delivery: T2: 1 month after delivery. Reported data are shown as dot plots and means ± 1SD of 5 different measurements. Square brackets indicate significance between groups (*p* < 0.05). Other significances are shown through symbols.

**Table 1 ijms-26-04962-t001:** Demographic variables of enrolled patients.

	Severe PE (n = 18)	Mild PE (n = 18)	HCs (n = 17)	P#	P£	P$
Mean age	33.82	33.14	30.40	0.820	1.000	0.710
Range	(27–41)	(25–47)	(21–42)			
SD	4391	6562	8081			
Age < 35	10/18	13/18	11/17	1.000	0.527	0.553
35–40	6/18	3/18	5/17			
>40	2/18	2/18	1/17			
Mean BMI	26.16	31.50	25.54	0.863	0.55	0.091
Range	(17.6–41)	(21.7–42.6)	(23.4–27.7)			
SD	6175	6364	1902			
BMI < 25	8/18	3/18	4/17	0.400	0.797	0.020
25–30	4/18	5/18	8/17			
>30	6/18	10/18	5/17			
Ethnicity: Caucasian	16/18	15/18	15/17	0.624	0.660	0.643
African	2/18	1/18	1/17			
Asian	0/18	2/18	1/17			
Smoking:						
Current	1/18	1/18	1/17	1.000	0.017	0.170
Previous	5/18	0/18	0/17			
Absent	12/18	17/18	16/17			

Legend. HCs = Healthy Controls; BMI = Body Mass Index; P# = Mild PE vs. Healthy controls; P£ = Severe PE vs. Healthy controls; P$ = Severe PE vs. Mild PE. *p* < 0.05 was considered statistically significant.

**Table 2 ijms-26-04962-t002:** Data on delivery and newborns.

		Severe PE (n = 18)	Mild PE (n = 18)	HCs (n = 17)	P#	P£	P$
Delivery (weeks)		33.39	37.39	37.47			
Range		(26–39)	(35–40)	(24–41)			
SD		3773	1339	4354			
Weeks	<32	7/18	0/18	2/17	0.890	0.004	0.002
	32–37	8/18	9/18	4/17			
	>37	3/18	9/18	11/17			
HELLP		5/18	0	0		<0.0001	<0.0001
Live newborns		100%	100%	100%			
Mean weight	<1000 g	3/18	0/18	0/17	0.076	0.002	<0.001
	1000–2500 g	11/18	2/18	5/17			
	>2500 g	4/18	16/18	12/17			
APGAR 1 min	7	0/18	1/18	0/17	0.371	0.003	0.090
	8	4/18	0/18	0/17			
	9	7/18	5/18	3/17			
	10	7/18	12/18	14/17			
NICU hospitalization	Yes	12/18	3/18	2/17	0.689	<0.001	0.002
	No	6/18	15/18	15/17			
Modality	CC	17/18	7/18	7/17	0.335	0.006	0.002
	ID	1/18	5/18	1/17			
	SP	0/18	6/18	9/17			

Legend. NICU = Neonatal Intensive Care Unit; CC: cesarean section; ID: induced delivery; SP: spontaneous delivery; HELLP: Hemolysis, Elevated liver enzymes, Low platelets; APGAR: Appearance, Pulse, Grimace, Activity, Respiration; P# = Mild PE vs. Healthy controls; P£ = Severe PE vs. Healthy controls; P$ = Severe PE vs. Mild PE. *p* < 0.05 was considered statistically significant.

**Table 3 ijms-26-04962-t003:** Clinical and biochemical variables in PE patients at diagnosis (T0) and in HCs.

		Severe PE (n = 18)	Mild PE (n = 18)	HCs (n = 17)	P#	P£	P$
GE (weeks)	<32	9/18	3/18	2/17	0.240	0.041	0.023
	32–37	7/18	9/18	4/17			
	>37	2/18	6/18	11/17			
AST (U/L)	<20	4/18	9/18	12/17	0.164	<0.001	0.010
	20–100	7/18	9/18	5/17			
	>100	7/18	0/18	0/17			
ALT (U/L)	<20	5/18	12/18	14/17	0.681	<0.001	0.001
	20–100	6/18	6/18	3/17			
	>100	7/18	0/18	0/17			
PLTs	<100 × 10^3^	3/18	0/18	0/17	0.073	0.005	0.065
	100–50 × 10^3^	5/18	3/18	0/17			
	>150 × 10^3^	10/18	15/18	17/17			
Uric acid (mg/dL)	<6	6/18	13/18	17/17	0.108	0.068	0.693
	>6	12/18	5/18	0/17			
Hb (g/dL)	<10.5	2/18	3/18	9/17	0.116	0.001	0.232
	10.5–12.5	7/18	10/18	7/17			
	>12.5	9/18	5/18	1/17			
SAP (mmHg)	<140	3/18	9/18	17/17	0.016	0.006	0.525
	140–160	7/18	7/18	0/17			
	>160	8/18	2/18	0/17			
DAP (mmHg)	<90	4/18	10/18	17/17	0.002	0.022	0.418
	90–110	11/18	5/18	0/17			
	>110	3/18	1/18	0/17			
Proteinuria (mg/day) SD		1589	1371	/	/	/	0.490
		1415	1152	/			
sFlt-1 (pg/mL)		5992	3902	/	/	/	0.0008
		1386	903				

Legend. GE = gestational age; AST = aspartate transaminase; ALT = alanine transaminase; Hb = hemoglobin; PTLs = platelets; SAP = Systolic arterial blood pressure; DAP = Diastolic arterial blood pressure; sFlt-1: Soluble fms-like tyrosine kinase-1; P# = Mild PE vs. HCs; P£ = Severe PE vs. HCs; P$ = Severe PE vs. Mild PE. *p* < 0.05 was considered statistically significant.

**Table 4 ijms-26-04962-t004:** Clinical and biochemical variables in PE patients at delivery (T1) and in HCs.

		Severe PE (n = 18)	Mild PE (n = 18)	HCs (n = 17)	P#	P£	P$
AST (U/L)	<20	4/18	8/18	12/17	0.081	<0.001	0.018
	20–100	8/18	10/18	5/17			
	>100	6/18	0/18	0/17			
ALT (U/L)	<20	5/18	12/18	14/17	0.431	<0.001	0.002
	20–100	6/18	6/18	3/17			
	>100	7/18	0/18	0/17			
PTLs	<100 × 10^3^	3/18	0/18	0/17	0.154	0.005	0.031
	100–150 × 10^3^	5/18	3/18	0/17			
	>150 × 10^3^	10/18	15/18	17/17			
Uric acid (mg/dL)	<6	6/18	10/18	17/17	0.116	0.272	0.846
	>6	12/18	8/18	0/17			
Hb (g/dL)	<10.5	9/18	9/18	9/17	0.894	0.469	0.412
	10.5–12.5	6/18	8/18	7/17			
	>12.5	3/18	1/18	1/17			
SAP (mmHg)	<140	4/18	10/18	17/17	0.200	0.079	0.485
	140–160	8/18	8/18	0/17			
	>160	6/18	0/18	0/17			
DAP (mmHg)	<90	4/18	9/18	17/17	0.870	0.055	0.152
	90–110	10/18	9/18	0/17			
	>110	4/18	0/18	0/17			

Legend. AST: aspartate transaminase; ALT: alanine transaminase; Hb = hemoglobin; PTLs = platelets; PAS = Systolic arterial blood pressure; PAD = Diastolic arterial blood pressure; P# = Mild PE vs. Healthy controls; P£ = Severe PE vs. Healthy controls; P$ = Severe PE vs. Mild PE. *p* < 0.05 was considered statistically significant.

## Data Availability

All clinical and laboratory data of patients enrolled in the study are available in the database of the “Maggiore della Carità” University Hospital in Novara, Italy. All data generated from EV analysis and from in vitro experiments performed on human GEC and PODO are available in the data center of the Aging Project of Excellence of the Department of Translational Medicine (DIMET), University of Piemonte Orientale (UPO). The datasets used and/or analyzed during the current study are available from the corresponding author on reasonable request.

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
