# Peer review of "Plasma Extracellular Vesicles from Preeclamptic Patients Trigger a Detrimental Crosstalk Between Glomerular Endothelial Cells and Podocytes Involving Endothelin-1"

_ijms, 2025, doi:10.3390/ijms26114962_

Round 1

Reviewer 1 Report

Comments and Suggestions for Authors

The manuscript ijms-3489804 “Plasma extracellular vesicles from preeclamptic patients trigger a detrimental crosstalk between glomerular endothelial cells and podocytes involving Endothelin-1” by Grossini et al. reports an interesting study that investigated the role of plasma extracellular vesicles in glomerular damage associated with preeclampsia.

The study was conducted through: 1) the demographic, clinical and biochemical profile of patients with preeclampsia; 2) the characterisation of extracellular vesicles by investigating the expression of lymphocyte, endothelial and platelet markers 3) the in vitro stimulation of glomerular endothelial cells and podocytes with plasma extracellular vesicles; 4) the glomerular endothelial cell-podocyte co-cultures in which one cell type was stimulated with the supernatant of the other cell line.

The authors conclude that plasma extracellular vesicles are involved in triggering the deleterious crosstalk between glomerular endothelial cells and podocytes that leads to the glomerular damage associated with preeclampsia, and that the study of plasma extracellular vesicles is a promising tool for a better understanding of the pathophysiology of preeclampsia.

The article is well structured, with satisfactory results supported by figures, clearly argued discussion, but the methods are poorly described and should be rewritten. Therefore, the article cannot be published in its current form, but needs major revision.

Below are suggestions for improving the quality of the article.

Introduction

Line 68: what does STB mean?

Results

Table 1: adjust the alignment of the lines in the ‘Smoking’ part.

Table 2: specify the meaning of HELLP and APGAR in the legend.

Line 99: write ‘VS’ in lower case.

Table 3: specify the meaning of sFlt-1 in the legend.

Lines 132, 157, 172, 195, 208,236: is “1SD” correct?

Lines 136-142: capitalise the letters in Figure 2.

Line 135-141: write the biomarkers used to detect T and B linfocytes, monocytes, endothelial cells, and platelets.

Lines 162-168: capitalise the letters in Figure 3.

Lines 176-189: capitalise the letters in Figure 4.

Lines 199-204: capitalise the letters in Figure 5.

Lines 216-220: capitalise the letters in Figure 6.

Material and Methods

Major Points

As said above, the Materials and Methods section has been described very superficially and needs to be rewritten. In particular, Section 4.3 In vitro studies is very lacking in details. It is necessary to describe in detail 1) how GEC and PODO were isolated and characterised; 2) how the direct stimulation with plasma-derived EVs of GEC and PODO was carried out; 3) what the GEC-PODO co-culture model devoloped by authors consists of; 4) the cell density of the GEC-PODO cultures and the volumes/concentration of supernatant produced by each cell line used to stimulate the other cell line.

Data on T- and B-lymphocytes, monocytes, endothelial cells and platelets were reported in the results. Write down which specific biomarkers were used and the methods used to analyse their expression.

Minor Points

Line 376: what does HELLP mean?

Line 390: indicate the source of EVs.

Line 393: what does PLAP and TF mean?

References

Write references according to the IJMS guideline.

Author Response

We thank the Reviewer for the suggestions aimed to improve the quality of our manuscript.

Comment:

Introduction

Line 68: what does STB mean?

Answer: We have now added syncytiotrophoblast (STB)

Comment:

Results

Table 1: adjust the alignment of the lines in the ‘Smoking’ part.

Answer: Thanks: done

Table 2: specify the meaning of HELLP and APGAR in the legend. 

Answer: done. HELLP: Hemolysis, Elevated liver enzymes, Low platelets; APGAR: Appearance, Pulse, Grimace, Activity, Respiration

Line 99: write ‘VS’ in lower case.

Answer: Thanks: done

Table 3: specify the meaning of sFlt-1 in the legend.

Answer: done. sFlt-1: Soluble fms-like tyrosine kinase-1

Lines 132, 157, 172, 195, 208,236: is “1SD” correct?

Answer: yes, it refers to 1 Standard Deviation (1SD) applied for the statistical test

Lines 136-142: capitalise the letters in Figure 2.

Answer: Thanks: done

Line 135-141: write the biomarkers used to detect T and B linfocytes, monocytes, endothelial cells, and platelets.

Answer: we have now reported in the figure legend the detailed description of biomarkers of T/B lymphocytes, monocytes and endothelial cells used for the experimental procedure.

Lines 162-168: capitalise the letters in Figure 3.

Answer: Thanks: done

Lines 176-189: capitalise the letters in Figure 4.

Answer: Thanks: done

Lines 199-204: capitalise the letters in Figure 5.

Answer: Thanks: done

Lines 216-220: capitalise the letters in Figure 6.

Answer: Thanks: done

Comment:

Material and Methods

Major Points: As said above, the Materials and Methods section has been described very superficially and needs to be rewritten. In particular, Section 4.3 In vitro studies is very lacking in details. It is necessary to describe in detail 1) how GEC and PODO were isolated and characterised; 2) how the direct stimulation with plasma-derived EVs of GEC and PODO was carried out; 3) what the GEC-PODO co-culture model devoloped by authors consists of; 4) the cell density of the GEC-PODO cultures and the volumes/concentration of supernatant produced by each cell line used to stimulate the other cell line.

Data on T- and B-lymphocytes, monocytes, endothelial cells and platelets were reported in the results. Write down which specific biomarkers were used and the methods used to analyse their expression.

Answer: we thank the Reviewer for this important suggestion. In the first version of the manuscript, we tried to limit the description of the methodology applied in the study in order to respect the word limit of the journal. We understand and fully agree that a short description of the assays may result in a difficult understanding of the processes. For this reason, we have now improved the description of the methods regarding EV isolation and phenotypic characterization including biomarkers of T/B lymphocytes, monocytes, endothelial cells and platelets, and we have also added a supplement file containing a detailed description of the in vitro assays performed on human GEC and PODO following plasma EV incubation. 

Minor Points

Line 376: what does HELLP mean?

Answer: HELLP: Hemolysis, Elevated liver enzymes, Low platelets

Line 390: indicate the source of EVs.

Answer: Thanks: done.

Line 393: what does PLAP and TF mean?

Answer: Placental Alkaline Phosphatase (PLAP) (Santa Cruz Biotechnologies, Santa Cruz, CA) or Tissue Factor (TF). Now added in the text.

Comment:

References

Write references according to the IJMS guideline.

Answer: we thank the Reviewer for the suggestion. We have now checked all the references in accordance with IJMS guidelines.

Reviewer 2 Report

Comments and Suggestions for Authors

dear authors,

 I read with great interest the manuscript, which falls within the aim of this Journal. In my honest opinion, the topic is interesting enough to attract the readers’ attention. Nevertheless, authors should clarify some points and improve the discussion, as suggested below. Authors should consider the following recommendations:

In my opinion you have to improve the paper refering in the text to the updated literature on this topic focusing how preeclampsia could determinate by many factors that has to be investigate carefully especially in case of in vitro fertilization conception and how ih the previous poliabortivity conception its really important to investigate in the infertile pathway all the possibile factors as thyroid disfuctions.

I also suggest to focus how its important to follow th neonatal Outcomes and Long-Term Follow-Up of Children Born from Frozen Embryo.

I suggest to read and cite these aticles:

 The Role of Cell and Gene Therapies in the Treatment of Infertility in Patients with Thyroid Autoimmunity
Fresh vs. frozen embryo transfer in assisted reproductive techniques: a single center retrospective cohort study and ethical-legal implications

Impact of assisted reproduction techniques on the neuro-psycho-motor outcome of newborns: a critical appraisa

The relationship between circulating endothelin-1, soluble fms-like tyrosine kinase-1 and soluble endoglin in preeclampsia

Author Response

Comment: In my opinion you have to improve the paper refering in the text to the updated literature on this topic focusing how preeclampsia could determinate by many factors that has to be investigate carefully especially in case of in vitro fertilization conception and how ih the previous poliabortivity conception its really important to investigate in the infertile pathway all the possibile factors as thyroid disfuctions. I also suggest to focus how its important to follow th neonatal Outcomes and Long-Term Follow-Up of Children Born from Frozen Embryo.I suggest to read and cite these aticles:

The Role of Cell and Gene Therapies in the Treatment of Infertility in Patients with Thyroid Autoimmunity
Fresh vs. frozen embryo transfer in assisted reproductive techniques: a single center retrospective cohort study and ethical-legal implications

Impact of assisted reproduction techniques on the neuro-psycho-motor outcome of newborns: a critical apprais

The relationship between circulating endothelin-1, soluble fms-like tyrosine kinase-1 and soluble endoglin in  preeclampsia

Answer: We thank the Reviewer for the relevant comments aimed to improve the quality of our work. We have now added in the manuscript and discussed the papers indicated by the Reviewer that are also cited in the References. 

Reviewer 3 Report

Comments and Suggestions for Authors

In this excellent manuscript the authors extend their previous investigations by conducting an in-depth time course analysis of the extracelluar vesicles (EVs) from the plasma of preeclamptic women and healthy women. It is well established that EVs may play a role in preeclampsia (PE)-associated endothelial dysfunction and glomerular damage. The focus of this study was to investigate the role of PE plasma EVs in the crosstalk between glomerular endothelial cells (GEC) and podocytes 34 (PODO) in their in vitro coculture model.

In the time course study clinical and laboratory variables were examined at T0 (diagnosis), T1 (delivery), T2 (one month after delivery) in 36 PE patients from both severe early onset and mild PE phenotypes and 17 age-matched controls.

Methods include NanoSight and MACSPlex to assess EV concentration, size, phenotype and source. In vitro GEC and PODO were stimulated with plasma EVs from each time point to study viability, reactive oxygen species (ROS) production, permeability to albumin, endothelial-to-mesenchymal transition, and Endothelin-1 release.

The results show that EV size and concentration were higher in PE (more in severe than in mild PE than in healthy controls and in severe than in mild forms of disease. The authors perform linear regression studies and show that at T0, higher EV concentration correlated with proteinuria, blood pressure, uric acid, liver enzyme levels and high levels of sFlt-1. In the flow cytometry marker analysis of EV origin they show that PE-EVs originated from leukocytes, endothelial cells, platelets, placenta, and they go on to show that they induced GEC and PODO damage demonstrated by reduction of viability, increased ROS release and albumin permeability.

Finally, and most interestingly using co-culture experiments they demonstrated that PE-EVs mediated a deleterious intraglomerular crosstalk via Endothelin-1 release from GEC which was able to down-regulate nephrin in PODO. They showed that this was specific by using an endothelin 1 receptor antagonist to block the effect. Interetignly although the EV effect is dimished by T2 q month after delivery some parameters are still abnormal as compared to controls leading the authors to suggest that this may be a continuing deleterious pathway that may contribute to the elevated cardiovascular risk associate with PE.

Q1) The authors mention the cardiovascular risk of PE a couple of time s in the discussion but do not reference this statement. Please do so.  

They conclude that PE plasma contains a peculiar pattern of EVs which is able to affect GEC and PODO functions and to induce proteinuria through Endothelin-1 involvement.

 The paper is well written and the study well planned and presented

Q2) My major concern is with the quality of the figures throughout the paper. These are particularly small and the text on axes and statistics is impossible to read and hard to make out the difference between the statistics and the data points. When magnified the image is blurry. Please ensure that the graphs are uploaded as high quality TIFFS 600-1200 dpi.

Q3) The authors suggest they will look at the contents of the EVS in future studies , however there are a number of peer reviewed reports of this already. Could the authors assess if any of the reported EV micro RNAs impact on the endothelin pathway.

Author Response

Comment: Q1) The authors mention the cardiovascular risk of PE a couple of times in the discussion but do not reference this statement. Please do so. 

Answer: We thank the Reviewer for this observation aimed to improve the quality of our manuscript. PE-associated cardiovascular risks are briefly discussed in the Introduction and Discussion. The most important point is that hypertension is a part of the PE classification and it obviously represents a risk factor for cardiovascular events. Moreover, a follow-up of PE patients in particular with reduced GFR levels is suggested in order to prevent progression toward chronic kidney disease (CKD) that in turn is correlated with an increased incidence of major adverse cardiovascular events (MACE). Last, the findings of the present study on plasma EV-induced GEC alterations may be expanded in other organs at the microcircualtory level. 

Comment: Q2) My major concern is with the quality of the figures throughout the paper. These are particularly small and the text on axes and statistics is impossible to read and hard to make out the difference between the statistics and the data points. When magnified the image is blurry. Please ensure that the graphs are uploaded as high quality TIFFS 600-1200 dpi.

Answer: We thank the Reviewer for underlying this relevant point. We have now improved the quality of the figures in accordance with these indications in order to facilitate the understanding of results.

Comment: Q3) The authors suggest they will look at the contents of the EVS in future studies , however there are a number of peer reviewed reports of this already. Could the authors assess if any of the reported EV micro RNAs impact on the endothelin pathway.

Answer: we agree with the Reviewer that the transfer of miRNA cargo from EV to target cells is probably the most important biological activity of these nanoparticles in the mechanisms of intercellular crosstalk. Our group has previosuly demonstrated that the regenerative potential of stem cell-derived EVs is mainly ascribed to the horizontal transfer of mRNAs and miRNAs to target cells (some of the papers are discussed and cited in the references). For this reason, we have now added in the last part of the Discussion some studies reporting the role of miRNAs in PE as both biomarkers and mediators of disease.

Reviewer 4 Report

Comments and Suggestions for Authors

The manuscript try to test a central hypothesis that extracellular vesicles phenotype can br related to PE patient severity. However the manuscript needs to be revised to be clearly understood.

Introduction is so short need to detail more the central problem with the current literature.

Methodology several problems were detected in this session, as above:

  • number of patients and groups was not detailed in the session 4.1
  • procedure for collection of plasma and other samples was not detailed
  • parameters used for EV classification is absent
  • origin of GEC and PODO cells was not detailed
  • in vitro studies procedures was not detailed, such as amount of cultured cells, media, amount of vesicles, how long cells were cultures with vesicles, etc.
  • only antibody brand is provided and also codes are absent, then is not possible to know the exact antibody was used

Results several points need to be improved to make results clear

  • as methodology is not well described is impossible to understand the origin of results. I.e. is not described in methodology how EV source was classified; cell culture parameters is absent such how cell viability, ROS and NO production was evaluated, in what time points.
  • several acronyms are not detailed
  • in not detailed how culture supernatants was collected, how is analyzed, which time point (or points) was used
  • figures low resolution make difficult to ready some graphs

Discussion as results is not completely clear (due absence of methodology details) is difficult to evaluated the discussion session

Conclusion this session is to accept or deny the hypothesis, however is presented much more as results summary. 

Author Response

Comment: Introduction is so short need to detail more the central problem with the current literature.

Answer: We thank the Reviewer for the comments aimed to improve the quality of the manuscript. In accordance with the suggestions of all the Reviewers, we implemented some parts of the Introduction. Unfortunately, the limited number of words requested by the journal and the amount of the data presented in the Results and discussed do not allow us to describe in more details other published papers in the field. However, we think that the previous literature and the aim of the study are clearly and adequately stated.

Comment: Methodology several problems were detected in this session, as above:

  • number of patients and groups was not detailed in the session 4.1
  • procedure for collection of plasma and other samples was not detailed
  • parameters used for EV classification is absent
  • origin of GEC and PODO cells was not detailed
  • in vitro studies procedures was not detailed, such as amount of cultured cells, media, amount of vesicles, how long cells were cultures with vesicles, etc.
  • only antibody brand is provided and also codes are absent, then is not possible to know the exact antibody was used

Answer: we thank the Reviewer for this important suggestion. In the first version of the manuscript, we tried to limit the description of the methodology applied in the study in order to respect the word limit of the journal. We understand and fully agree that a short description of the assays may result in a difficult understanding of the processes. For this reason, we have now improved the description of the methods regarding patient characteristics, EV isolation and phenotypic characterization including biomarkers of T/B lymphocytes, monocytes, endothelial cells and platelets, and we have also added a supplement file containing a detailed description of the in vitro assays performed on human GEC and PODO following plasma EV incubation.

Comment: Results several points need to be improved to make results clear

  • as methodology is not well described is impossible to understand the origin of results. I.e. is not described in methodology how EV source was classified; cell culture parameters is absent such how cell viability, ROS and NO production was evaluated, in what time points.
  • several acronyms are not detailed
  • in not detailed how culture supernatants was collected, how is analyzed, which time point (or points) was used
  • figures low resolution make difficult to ready some graphs

Answer: We thank the Reviewer for this comment. We hope that the changes made in the Methods will be useful to better understand the results presented in the paper. We have tried to better clarify acronyms in the Results section and in all figure legends. Moreover, in accordance with the request of other Reviewers, we improved the quality of the figures for better understanding the results. The other points about the Results section have already been further discussed for Methods above. 

Comment: Discussion as results is not completely clear (due absence of methodology details) is difficult to evaluated the discussion session. Conclusion this session is to accept or deny the hypothesis, however is presented much more as results summary. 

Answer: We thank the Reviewer for this comment. We hope that the changes made in the Methods and Results will be useful to better understand the Discussion and Conclusions of the paper. As observed by the other Reviewers, the discussion and the conclusions reported the main points of novelty of the study concerning the role of plasma EV in the determination of glomerular injury and development of proteinuria in PE patients with a translational approach that first correlated EV concentration and phenotype with lab tests and clinical data and then demonstrated in vitro on human GEC and PODO which are some new potential mechanisms of EV-induced maladaptive intraglomerular crosstalk involving Endothelin-1. Last, in the discussion and conclusions we also suggested how new therapeutic approaches such sa ET-1 inhibition or apheresis may have a potential role in PE. management. The other  points about the Discussion and Conclusions have already been further elucidated for Methods and Results above.

Round 2

Reviewer 1 Report

Comments and Suggestions for Authors

The manuscript has been corrected according to the suggestions of this referee. The paper can be accepted for publication. 

Reviewer 2 Report

Comments and Suggestions for Authors

Dear authors,

I read with great interest the manuscript, which falls within the aim of this Journal. In my honest opinion, the topic is interesting enough to attract readers’ attention. Nevertheless, the authors should clarify some points and improve the discussion, as suggested below. The authors should consider the following recommendations:

In my opinion, you have to improve the paper by referring if plasma extracellular vesicles from preeclamptic patients are different if pts with prevoius infection as in pandemia ERA as covid or zika as well and if it could related to Serum Laeverin Levels

I SUGGEST YOU READ and cite these articles:

Congenital Zika Syndrome: Genetic Avenues for Diagnosis and Therapy, Possible Management and Long-Term Outcomes

Amniotic Fluid and Maternal Serum Laeverin Levels and Their Correlations with Fetal Size and Placental Volume in Second Trimester of Pregnancy-A Prospective Cross-Sectional Study

Obstetric and Gynecological Admissions and Hospitalizations in an Italian Tertiary-Care Hospital during the COVID-19 Pandemic: A Retrospective Analysis According to Restrictive Measures

Comparison of Maternal and Neonatal Outcomes between SARS-CoV-2 Variants: A Retrospective, Monocentric Study

Preterm Birth and SARS-CoV-2: Does a Correlation Exist?